# Second Wave of the Study of Taiwanese Caregivers of Children with ADHD in the COVID-19 Pandemic: Intentions to Vaccinate Their Children for COVID-19, and Related Factors

**DOI:** 10.3390/vaccines10050753

**Published:** 2022-05-11

**Authors:** Ching-Shu Tsai, Liang-Jen Wang, Ray C. Hsiao, Cheng-Fang Yen

**Affiliations:** 1Department of Child and Adolescent Psychiatry, Chang Gung Memorial Hospital, Kaohsiung Medical Center, Kaohsiung 83301, Taiwan; jingshu@cgmh.org.tw (C.-S.T.); anus78@cgmh.org.tw (L.-J.W.); 2School of Medicine, Chang Gung University, Taoyuan 33302, Taiwan; 3Department of Psychiatry, Children’s Hospital, Seattle, WA 98105, USA; rhsiao@u.washington.edu; 4Department of Psychiatry and Behavioral Sciences, School of Medicine, University of Washington, Seattle, WA 98195, USA; 5Department of Psychiatry, Kaohsiung Medical University Hospital, Kaohsiung 80756, Taiwan; 6Department of Psychiatry, School of Medicine, College of Medicine, Kaohsiung Medical University, Kaohsiung 80708, Taiwan; 7College of Professional Studies, National Pingtung University of Science and Technology, Pingtung 91201, Taiwan

**Keywords:** ADHD, caregiver, COVID-19, vaccine

## Abstract

The second wave of the Study of Taiwanese Caregivers of Children with Attention-Deficit/Hyperactivity Disorder (ADHD) in the COVID-19 Pandemic was conducted at the time of a severe COVID-19 outbreak. The aims of this study were to compare the level of the intentions of caregivers of children with ADHD to vaccinate their children between the first and second waves of study, as well as to examine the COVID-19 pandemic and non-COVID-19 pandemic factors related to caregivers’ intentions. In total, 252 caregivers of children with ADHD completed the structured questionnaires, including the Drivers of COVID-19 Vaccination Acceptance Scale; the Risk Perception of the COVID-19 Scale; caregivers’ Difficulties in Asking Their Children to Adopt Self-Protective Behavior Scale; the Brief Symptom Rating Scale; the Parental Bonding Instrument; the Swanson, Nolan, and Pelham version IV Scale; and the questionnaires for the intentions to vaccinate their children and child’s medication use for treating ADHD. The results demonstrated that 82.5% of caregivers reported their willingness to vaccinate their children definitely or under doctors’ recommendation; the level of intentions to vaccinate significantly increased compared with that of caregivers in the first wave of the study. Caregivers’ drivers of COVID-19 vaccination uptake, namely, values, impact, and autonomy but not knowledge; being male caregivers; being caregivers of girls; and the older age of the children were positively associated with caregiverscaregivers’ intentions. The specific intervention programs for enhancing caregivers’ intentions should be specified according to the sex and age of caregivers and of the children with ADHD. The Drivers of COVID-19 Vaccination Uptake should be also the target of intervention for enhancing caregivers’ intentions through strengthening caregivers’ acceptance of the COVID-19 vaccines’ values, positive impact and autonomy to vaccinate their children.

## 1. Introduction

### 1.1. Coronavirus Disease in the General Population and Individuals with Mental Disorders

Coronavirus disease (COVID-19) caused by severe acute respiratory syndrome coronavirus 2 (SARS-CoV-2) has spread worldwide. As of 3 April 2022, the World Health Organization (WHO) had recorded 486,761,597 confirmed COVID-19 cases, including 6,142,735 deaths [1]. As a novel and highly contagious respiratory infectious disease, the COVID-19 pandemic has impacted people’s lives in many ways, including health [2,3,4], business activities [5], education and training [6], multiple aspects of lifestyles [7], working opportunity and time [8], and interpersonal interactions [9].

Meta-analysis studies have found that individuals with pre-existing mental disorders are at increased risk of COVID-19 mortality [10,11] and hospitalization [11]. The results of meta-analysis studies highlight the need for targeted approaches to manage and prevent COVID-19, for example, vaccination in individuals with mental disorders [10,11]. Along with psychotic and mood disorders, attention-deficit/hyperactivity disorder (ADHD) has been identified to predict the outcomes of contracting COVID-19. Compared with individuals without ADHD, those with ADHD have a higher risk of contracting respiratory infectious diseases (RIDs) [12], including COVID-19 [13,14]. Moreover, ADHD is a risk factor for poor COVID-19 outcomes (e.g., poor adaptation to the pandemic, severe COVID-19 symptoms, and increased referral for hospitalization) [15,16]. Caregivers of children with ADHD reported experiencing a heavy care burden, stressful feelings, mental health disturbances, and compromised quality of life during the COVID-19 pandemic [16,17,18].

### 1.2. Vaccines for COVID-19

Vaccines might stop the spread of and the death caused by COVID-19 [19]. A review and meta-analysis study reported that complete COVID-19 vaccination significantly reduced the risks of COVID-19 infection, hospitalization, admission to the intensive care unit, and death [20]. However, a study in Denmark found that vaccine willingness for preventing COVID-19 was lower amongst individuals with mental disorders, compared with the general population [21]. A study in China found that individuals with mental disorders had a much lower rate of vaccination than the general population [22]. These studies have suggested that it is necessary to develop strategies for increasing vaccination coverage among individuals with mental disorders [21,22].

Review studies have indicated that although the risk of severe acute COVID-19 in healthy children infected with SARS-CoV-2 is considerably lower than that in adults [23,24], protecting children from the long-term consequences of COVID-19 infection and quarantine and school closures during the COVID-19 pandemic is essential [25]. The BNT162b2 vaccine (Pfizer-BioNTech) [26] and the mRNA-1273 vaccine (Moderna) [27] have been observed to be effective at reducing the risk of COVID-19 in children aged 12–17 years and have been authorized for emergency use [28,29,30,31]. Studies examining the effectiveness and safety of COVID-19 vaccines in children aged < 12 years are ongoing [25].

Vaccination uptake for COVID-19 may play a role in reducing the risk of COVID-19 in children with ADHD and caregivers’ concerns regarding the adverse effects of COVID-19 on their children with ADHD. However, a hesitancy to vaccinate their children against COVID-19 is common among caregivers [17,32,33,34,35,36,37,38]. Factors related to the intentions of caregivers of children with ADHD to vaccinate their children should be determined.

### 1.3. COVID-19 Pandemic Factors Related to Caregivers’ Intentions to Vaccinate Their Children

Both COVID-19 pandemic (proximal) and non-COVID-19 pandemic (background) factors may be related to caregivers’ intentions to vaccinate their children against COVID-19. Caregivers’ cognitive drives to get vaccinated constitute the first COVID-19 pandemic factor that should be examined. According to the cognitive model of empowerment (CME) [39], caregivers’ vaccination intentions are driven by four distinct cognitive processes: (1) values (caregivers’ consideration of the purposes and values of vaccine uptake for COVID-19); (2) impact (caregivers’ belief that vaccine uptake can make a difference and achieve its purpose, that is, prevent COVID-19 infection and transmission and reduce the severity of COVID-19 symptoms); (3) knowledge (caregivers’ belief that they have adequate information and knowledge regarding vaccines against COVID-19); and (4) autonomy (caregivers’ belief that the initiation of vaccine uptake for COVID-19 is self-determined).

According to protection motivation theory [40], caregivers who perceived a higher risk of COVID-19 were more intent to vaccinate their children [36,37,41]. Moreover, according to the health belief model (HBM) [42], individuals who have high confidence in coping with COVID-19 may successfully adopt and persuade others to adopt effective strategies for preventing the contraction of COVID-19. Caregivers may have high intentions to vaccinate their children with ADHD and expect the protecting effects of vaccination from COVID-19 for their children if they have the difficulty in asking their children to adopt self-protective behaviors. A previous study found that nearly a quarter of caregivers of children with ADHD suffered from psychological distress in the COVID-19 pandemic [18]. According to the HBM [42], caregivers’ decisions to vaccinate their children involve a complex process consisting of the cognitive, affective, and behavioral dimensions of their health-related beliefs. The poor mental health of caregivers may be detrimental to their intentions to vaccinate their children with ADHD. However, the roles of risk perception, difficulty in asking their children to adopt self-protective behaviors, and mental health status in caregivers’ intentions to vaccinate their children with ADHD merit further investigation.

### 1.4. Non-COVID-19 Pandemic Factors Related to Caregivers’ Intentions to Vaccinate Their Child

Non-COVID-19 pandemic factors that may be related to caregivers’ intentions to vaccinate their children against COVID-19 are divided into ADHD-related (receiving medication for ADHD and ADHD and comorbid oppositional defiant disorder (ODD) symptoms) and ADHD-unrelated factors (demographics and parenting styles). Untreated ADHD is a risk factor for COVID-19 infection, and medication can ameliorate this effect [14]. Children who do not receive medication for ADHD and have severe ADHD and ODD symptoms may experience difficulties in cooperating with caregivers’ instructions for adopting self-protective behaviors [13,16]. Caregivers may believe that vaccination would protect children who have difficulty in adopting self-protective behaviors from COVID-19. Regarding ADHD-unrelated factors, a child’s older age [32,34], caregivers’ higher education level [33,34,35], and caregivers’ male sex [34] have been reported to be related to caregivers’ high intentions to vaccinate their children. However, no study has examined the association between caregivers’ parenting styles and intentions to vaccinate their children. Parenting styles reflect caregivers’ attitudes and behaviors toward their children in early developmental stages [43]. Affectional or caring parenting represents warm and caring parenting behaviors, whereas overprotective parenting reflects the denial of the psychological autonomy of children [44,45]. Caregivers with affectional or caring parenting may vaccinate their children to express their care, whereas caregivers with overprotective parenting may vaccinate their children to control them and reduce COVID-19 risk. The relationship between the non-COVID-19 pandemic factors and caregivers’ intentions to vaccinate their children warrants further study.

### 1.5. Study of Taiwanese Caregivers of Children with ADHD in the COVID-19 Pandemic

The first COVID-19 case in Taiwan was identified on 29 January 2020. Because of the early closure of borders and the adoption of proactive containment and comprehensive contact tracing [46], Taiwan had a lower number of COVID-19 cases than numerous other countries or regions before May 2021. We conducted the first wave of the Study of Taiwanese Caregivers of Children with ADHD in the COVID-19 Pandemic between October 2020 and April 2021. The findings of the first wave of the study indicated that 23.0% of caregivers reported a willingness to let their children take vaccine shots against COVID-19 if the vaccines were available, 39.8% reported that they would let their children take vaccine shots under their doctors’ recommendation, 25.5% felt hesitant about vaccinating their children, and 11.8% explicitly refused to vaccinate their children [35]; caregivers’ concerns regarding the safety of vaccines, children’s regular use of medication for ADHD, and less severe ODD problems were negatively associated with caregivers’ intentions [47]. However, Taiwan experienced a severe COVID-19 outbreak between May and July 2021, during which time schools were closed for the first time to prevent the spread of COVID-19. Children were asked to stay at home to reduce their risk of COVID-19. Because of being forced to work at home or to stop working, many caregivers had more time to interact with their children with ADHD during this period compared with before the pandemic. We conducted the second wave of the Study of Taiwanese Caregivers of Children with ADHD in the COVID-19 Pandemic between August 2021 and January 2022 due to the following reasons. First, vaccines against COVID-19 have been available in Taiwan since April 2021, and caregivers’ intentions to vaccinate their children with ADHD might change due to the availability of vaccines and the outbreak of COVID-19. Second, the associations of several COVID-19 pandemic (e.g., caregivers’ drivers of accepting COVID-19 vaccines, risk perception, difficulty in managing their children’s self-protective behaviors, and mental health status in the pandemic) and non-COVID-19 pandemic factors (e.g., parenting styles) with caregivers’ intentions to vaccinate their children against COVID-19 were not examined in the first wave of study.

### 1.6. Study Aims

This study had two aims. First, we compared the level of the intentions of caregivers of children with ADHD to vaccinate their children between the first and second waves of study. Second, we examined the COVID-19 pandemic and non-COVID-19 pandemic factors related to caregivers’ intentions to vaccinate their children with ADHD against COVID-19. We hypothesized that caregivers of children with ADHD were more intent to vaccinate their children against COVID-19 in the second wave of study than in the first wave of study. Second, we hypothesized that the COVID-19 pandemic (caregivers’ drivers of accepting COVID-19 vaccines, risk perception, difficulty in managing their children’s self-protective behaviors, and mental health status in the pandemic) and non-COVID-19 pandemic factors (demographics, children’s medication use for treating ADHD, parenting styles, and ADHD and ODD symptoms) would be significantly related to caregivers’ intentions to vaccinate their children with ADHD against COVID-19.

## 2. Methods

### 2.1. Participants

Caregivers of children with ADHD who were aged 6–18 years that were visiting the child psychiatric outpatient clinics of two medical centers in Kaohsiung, Taiwan and were diagnosed as having ADHD according to the *Diagnostic and Statistical Manual of Mental Disorders, Fifth Edition* [48] were invited to participate in the present study between August 2021 and January 2022. Caregivers who had any cognitive impairment (e.g., addictive substance use, schizophrenia, and intellectual disability) that could have prevented them from understanding the purpose and procedure of the study were excluded. Informed consent was obtained from all the participants prior to the assessment conducted in the present study. The participants completed research questionnaires that were used to evaluate caregivers’ intentions to vaccinate their children against COVID-19 and COVID-19 pandemic and non-COVID-19 pandemic factors. The participants were allowed to seek help from research assistants when they experienced difficulty in completing the questionnaire. In total, 252 caregivers of children with ADHD participated in this study and completed self-report research questionnaires. The present study was approved by the Institutional Review Boards of Chang Gung Medical Foundation (202002118B0C501) and the Kaohsiung Medical University Hospital (KMUHIRB-E(I) 20200408).

### 2.2. Measures

#### 2.2.1. Intentions to Vaccinate Their Children against COVID-19

We used two items to evaluate caregivers’ intentions to vaccinate their children with ADHD against COVID-19: “When a COVID-19 vaccine becomes available, will you be willing to vaccinate your child?” and “Please rate your current willingness to let your child receive a COVID-19 vaccine.” The responses to the first item were “definitely not willing”, “not sure”, “if my doctor recommends it, I would let my child receive it”, and “definitely willing”. The second item was rated on a Likert scale from 1 (*very low*) to 10 (*very high*) [49].

#### 2.2.2. COVID-19 Pandemic Factors

##### Drivers of COVID-19 Vaccination Acceptance Scale

The Drivers of COVID-19 Vaccination Acceptance Scale (DrVac-COVID19S) [50] consists of 12 items that are rated on a 7-point Likert scale from 1 (*strongly disagree*) to 7 (*strongly agree*). These 12 items correspond to the CME model [39]: values (e.g., “It is important that I get the COVID-19 jab”), impact (e.g., “Vaccination is very effective in protecting me from COVID-19”), knowledge (e.g., “I know very well how vaccination protects me from COVID-19”), and autonomy (e.g., “I can choose whether to get a COVID-19 jab”). Each domain contains three items; all the three items are added up to obtain a total domain score. A higher score indicates a higher acceptance of the COVID-19 vaccine. A study supported the use of the four-factor structure model (i.e., using four CME constructs) of the DrVac-COVID19S [50]. In addition, the known-group validity of the DrVac-COVID19S is satisfactory; the measurement invariance of the DrVac-COVID19S across the subgroups of various sexes has been indicated [50,51]. Cronbach’s α in the present study was 0.86.

##### Risk Perception of the COVID-19 Scale

We used the five-item Risk Perception of COVID-19 Scale [52] to measure the risk perception of COVID-19. The questionnaire examined respondents’ levels of worry regarding contracting COVID-19 (1 item, rated on a 5-point Likert scale), worry regarding the general COVID-19 situation in the world (1 item, rated on a 10-point Likert scale), and worry regarding the severity of symptoms caused by COVID-19 (1 item, rated on a 5-point Likert scale), and their perceived likelihood of contracting COVID-19 (2 item, rated on a 7-point Likert scale). A higher total score indicates higher risk perception. The Risk Perception of COVID-19 Scale has acceptable internal consistency and concurrent validity for self-protective behaviors against COVID-19 in the general population in Taiwan [53,54]. Cronbach’s α in the present study was 0.70.

##### Caregivers’ Difficulties in Asking Their Children to Adopt Self-Protective Behavior Scale

We used a six-item scale to determine the frequency of difficulties encountered by caregivers in asking their children to adopt protective behaviors against COVID-19 [18]. The scale was developed on the basis of the recommendations of the Centers for Disease Control and Prevention for protecting against COVID-19 in the general population, including washing hands frequently, wearing a mask at all times, avoiding visiting crowded places, and practicing social distancing [55]. This scale contains other items specific to children: not touching their mouth, nose, objects, or other people in public places. Each item was rated on a 4-point scale from 0 (*never*) to 3 (*often*). A total score indicates the level of difficulty encountered by caregivers in asking their children to adopt protective behaviors against COVID-19. A study reported that the scale demonstrated high internal consistency and concurrent validity with caregivers’ mental health problems [18]. The Cronbach’s α value in this study was 0.842.

##### Brief Symptom Rating Scale

We used the five-item Brief Symptom Rating Scale (BSRS-5) to assess caregivers’ anxiety, depression, hostility, interpersonal hypersensitivity, inferiority, and insomnia during the COVID-19 pandemic [56]. Each item of the scale is rated on a 5-point scale from 0 (*not at all*) to 4 (*extremely*); the score of each item is added up to obtain a total score. A study indicated that a total BSRS-5 score of ≥6 can differentiate between individuals with and without psychiatric diagnosis [56]. Therefore, caregivers with a total BSRS-5 score of ≥6 were classified as having poor mental health.

#### 2.2.3. Non-COVID-19 Pandemic Factors

##### Sociodemographic Characteristics

Information on caregivers’ sex (0 = female; 1 = male), age, and education level, and on children’s sex (0 = girl; 1 = boy) and age, was collected.

##### Medication Use among Children for Treating ADHD

One question was used to determine the frequency of medication prescribed by doctors to the child for ADHD treatment [18]. The item was scored on a scale of 0 (*never received medication*) to 3 (*always regularly received medication*). The children of caregivers who scored 3 were considered to regularly receive medication for ADHD, whereas those of caregivers who scored <3 were considered to not regularly receive medication for ADHD.

##### Parental Bonding Instrument

We used the traditional Chinese version [57,58] of the Parental Bonding Instrument (PBI)—Parent Version [59] to measure parenting behaviors in three dimensions, namely, parental affection/care (12 items, e.g., “I could make the child feel better when he or she was upset.”), parental overprotection (seven items, e.g., “I tried to control everything that the child did.”), and authoritarian parenting (six items, e.g., “I let the child do things that he or she liked to do.”). Caregivers rated each item on a 4-point Likert scale ranging from 1 (*agree*) to 4 (*disagree*). We reverse-coded the items to facilitate the interpretation process. Caregivers with a high score in the parental affection/care dimension were considered to be highly affectionate and warm, those with a high score in the authoritarian parenting dimension were considered to highly encourage behavioral freedom, and those with a high score in the parental overprotection dimension were considered to have a high degree of denial of psychological autonomy. Studies have indicated that the traditional Chinese version of the PBI has satisfactory reliability and validity [58]. The Cronbach’s α values of the affection/care, overprotection, and authoritarian parenting dimensions in this study were 0.82, 074, and 0.71, respectively.

##### Swanson, Nolan, and Pelham, Version IV Scale

The ADHD and ODD symptoms of the children in the month preceding the commencement of the present study were examined using the traditional Chinese version [60] of the Swanson, Nolan, and Pelham, Version IV (SNAP-IV) Scale [61]. Caregivers rated the items of the subscales of inattention (9 items), hyperactivity/impulsivity (9 items), and ODD symptoms (8 items) on a 4-point Likert scale ranging from 0 (*not at all*) to 3 (*extremely*). The Cronbach’s α values of the inattention, hyperactivity/impulsivity, and ODD subscales in the present study were 0.89, 0.90, and 0.92, respectively.

### 2.3. Statistical Analysis

Statistical analyses were performed using SPSS 24.0 (SPSS Inc., Chicago, IL, USA). Results regarding caregivers’ intentions to vaccinate their children and the related COVID-19 pandemic and non-COVID-19 pandemic factors are presented as percentages and means with standard deviations (SDs). We examined the skewness and kurtosis of continuous variables and determined that all of their absolute values were <2, indicating normal distributions [62]. The associations between caregivers’ level of intentions to vaccinate their children against COVID-19 (dependent variable) and the COVID-19 pandemic (caregivers’ drivers of accepting COVID-19 vaccines, risk perception, difficulty in managing their children’s self-protective behaviors, and mental health status in the pandemic) and non-COVID-19 pandemic factors (demographics, children’s medication use for treating ADHD, parenting styles, and ADHD and ODD symptoms) (independent variables) were first examined by performing univariate regression analysis. The factors that were significantly associated with caregivers’ level of intentions to vaccinate their children were further entered into the multivariate regression analysis. A stepwise multivariate regression analysis model was used to prevent collinearity. A two-tailed *p* value of <0.05 indicated statistical significance.

## 3. Results

Table 1 presents the association between caregivers’ intentions to vaccinate their children and the COVID-19 pandemic and non-COVID-19 pandemic factors. Among the 252 caregivers, 200 were women and 52 were men. The mean age of caregivers was 42.23 (SD = 8.34) years, and their mean years of education was 14.17 (SD = 2.67) years. Furthermore, 91 (36.1%) reported poor mental health status. Among the 252 children with ADHD, 200 were boys and 52 were girls. The mean age of the children was 9.61 (SD = 2.39) years, and 212 (84.1%) received medication for ADHD regularly. The mean scores of inattention, hyperactivity/impulsivity, and ODD symptoms were 12.88 (SD = 5.83), 9.93 (SD = 6.17), and 9.33 (SD = 5.92), respectively. The mean scores of the value, impact, knowledge, and autonomy dimensions of the DrVac-COVID19S were 17.92 (SD = 2.81), 16.88 (SD = 2.91), 14.88 (SD = 3.00), and 14.00 (SD = 3.56), respectively. The mean scores of caregivers’ risk perception regarding COVID-19 and difficulty in managing their children’s self-protective behaviors were 14.02 (SD = 5.19) and 3.87 (SD = 3.56), respectively. The mean scores of the affection/care, overprotection, and authoritarian dimensions of the PBI were 37.08 (SD = 5.16), 13.75 (SD = 3.32), and 12.24 (SD = 2.67), respectively.

Regarding caregivers’ intentions to vaccinate their children, 91 (36.1%) caregivers reported the willingness to let their child take vaccine shots against COVID-19, 117 (46.4%) would let their child take vaccine shots under their doctors’ recommendation, 40 (15.9%) felt hesitant about vaccination, and 4 (1.6%) definitely refused to vaccinate their children. The proportion of caregivers who were willing to vaccinate their children definitely or under doctors’ recommendation increased from 62.8% in the first wave of the study to 82.5% in the second wave of the study (*p* < 0.001); the mean level of intentions to vaccinate their children increased from 6.08 (SD = 2.85) in the first wave of the study to 7.85 (SD = 2.23) in the second wave of the study (*p* < 0.001).

Table 2 presents the results of univariate and multivariate regression analyses performed to examine the COVID-19 pandemic and non-COVID-19 pandemic factors related to caregivers’ intentions to let their child receive vaccine shots. The results of the univariate regression analysis demonstrated that all four dimensions of caregivers’ drivers of COVID-19 vaccination uptake were positively associated with caregivers’ intentions to let their child receive vaccine shots. Male caregivers had a higher level of intentions than did female caregivers. Caregivers of the female children had a higher level of intentions than did caregivers of the male children. The older ages of both caregivers and children were positively associated with caregivers’ intentions to vaccinate. Caregivers’ risk perception, difficulty in monitoring their children’s adoption of self-protective behaviors; caregivers’ mental health status; parenting styles; and children’s medication use for treating ADHD; and children’s ADHD and ODD symptoms were not significantly associated with caregivers’ intentions to vaccinate.

The results of stepwise multivariate regression analysis revealed that caregivers’ drivers of COVID-19 vaccination uptake, namely, values, impact, and autonomy but not knowledge, were positively associated with caregivers’ intentions to let their children receive vaccine shots, whereas comorbid ODD symptoms were positively associated with caregivers’ intentions to vaccinate. Being male caregivers, being caregivers of girls, and the older age of the children were positively associated with caregivers’ intentions to vaccinate.

## 4. Discussion

The results of the second wave of the study demonstrated that 82.5% of caregivers who recently experienced a severe COVID-19 outbreak reported their willingness to vaccinate their children with ADHD against COVID-19 definitely or under doctors’ recommendation; the level of intentions to vaccinate significantly increased compared with that of caregivers in the first wave of the study during the remission of the pandemic. The present study demonstrated that caregivers’ drivers of COVID-19 vaccination uptake, namely, values, impact, and autonomy but not knowledge, being male caregivers, being caregivers of girls, and older age of the children were positively associated with caregivers’ intentions to vaccinate their children with ADHD.

Individuals in Taiwan experienced the severe effects of the COVID-19 pandemic for the first time between May and July of 2021. Thus, caregivers would have considered that vaccination can protect their children from contracting COVID-19, quarantine, and school closures during the COVID-19 pandemic. Moreover, the first COVID-19 vaccine (Oxford/AstraZeneca COVID-19 vaccine) has been available for use among adults in Taiwan since April 2021. Although the BNT162b2 vaccine was not available for children aged between 12 and 18 years in Taiwan until September 2021 and no vaccine is yet available for children aged <12 years, caregivers might have received information regarding COVID-19 in the second wave of the study but not in the first wave of the study; this information would have enhanced their intentions to vaccinate their children. However, 17.5% of caregivers were still hesitant or refused to vaccinate their children with ADHD. Concerns regarding COVID-19 vaccines should be understood; further interventions are required to improve caregivers’ acceptance of COVID-19 vaccines for their children.

The present study demonstrated that caregivers’ drivers of COVID-19 vaccine uptake, namely, values, impact, and autonomy, were positively associated with the intentions to vaccinate their children with ADHD. The results expand the CME by indicating that caregivers’ drivers of vaccine uptake may empower their intentions to vaccinate their children. The results of this study indicated that intervention programs should strengthen caregivers’ beliefs in the value of protecting their children from COVID-19 and the positive impact of COVID-19 vaccines on children’s lives. In addition, intervention programs must respect caregivers’ autonomy regarding vaccinating their children and help them communicate with family members who oppose vaccinating children. Caregivers’ knowledge regarding COVID-19 vaccines was not significantly associated with the intentions to vaccinate their children with ADHD. COVID-19 vaccines were developed in a short period; thus, individuals may not be familiar with mechanisms through which vaccines protect people from contracting COVID-19, especially in children. Caregivers may evaluate the necessity of vaccinating their children considering other reasons, for example, their beliefs in the values of COVID-19 vaccines, subjective norms [63], and confidence in the policy of government [64] but not their beliefs that they have sufficient knowledge regarding COVID-19 vaccines.

The present study demonstrated that caregivers’ male sex and children’s older age were positively associated with caregivers’ intentions to vaccinate their children with ADHD; these findings are similar to those of previous studies on caregivers of children in community [32,33,34,35]. The first wave of the Study of Taiwanese Caregivers of Children with ADHD in the COVID-19 Pandemic indicated that children’s regular use of medication for ADHD was negatively associated with caregivers’ intentions to vaccinate, whereas the children’s comorbid ODD symptoms were positively associated with caregivers’ intentions to vaccinate [47]; however, the present study did not demonstrate the same results. Moreover, in contrast to our hypotheses, the present study did not indicate a significant association between caregivers’ intentions to vaccinate and their risk perception, difficulty in managing their children’s self-protective behaviors, mental health status, and parenting styles. Additional studies are required to validate the results. Despite lacking significant associations with caregivers’ intentions to vaccinate their children with ADHD, caregivers’ mental health and difficulty in managing their children’s self-protective behaviors are concerning and thus warrant attention from health professionals.

This study focused on the intentions to vaccinate children against COVID-19 and related factors in caregivers of children with ADHD. The results of this study provided the references for developing intervention programs to enhance caregivers’ intentions. Given that individuals with mental disorders are the group at the risk of contracting COVID-19 and poor outcomes [28,29], further study is needed to examine the real rate of vaccination, vaccine willingness, barriers to assess vaccination, and effects of vaccination for COVID-19 in individuals with mental disorders during the COVID-19 pandemic. However, this study has several limitations. First, the participants were enrolled from pediatric psychiatric outpatient clinics. Thus, the results of this study cannot be generalized to caregivers who do not seek medical assistance for treating their children with ADHD. Second, we obtained all our data from only caregivers, which could have resulted in the problem of shared-method variance due to the use of a single information source. Third, caregivers of children without ADHD were not recruited into this study for comparison. Further studies are warranted to examine whether the factors related to caregivers’ intentions identified in this study also relate to the intentions of caregivers of children without ADHD.

## 5. Conclusions

This study indicated that although caregivers who experienced a severe COVID-19 outbreak had a higher level of intentions to let their child with ADHD receive vaccine shots compared with those who only experienced a mild COVID-19 outbreak, 17.5% of caregivers were hesitant or had no willing to let their child receive vaccine shots after the severe outbreak; their worry regarding COVID-19 vaccines should be understood. Furthermore, caregivers’ drivers of COVID-19 vaccine uptake regarding values, impact, and autonomy were positively associated with their intentions to let their children with ADHD receive vaccine shots. We suggest that caregivers who feel hesitant or have no willing to let their children receive vaccine shots require specific intervention programs to enhance their intentions. Factors related to caregivers’ intentions found in this study should be considered for developing intervention programs. As a modifiable factor, caregivers’ drivers of COVID-19 vaccination uptake should be the target of intervention. Intervention programs for enhancing caregivers’ intentions should be specified according to the sex and age of caregivers and children.

## Figures and Tables

**Table 1 vaccines-10-00753-t001:** The association between caregivers’ intentions to vaccinate their children and COVID-19 pandemic and non-COVID-19 pandemic factors (N = 252).

	n (%)	Mean (SD)	Range
*Caregivers’ intentions to vaccinate their children*			
Category variable			
Definitely willing	91 (36.1)		
Willing if doctors recommend	117 (46.4)		
Not sure	40 (15.9)		
Definitely not willing	4 (1.6)		
Level of intentions to vaccinate		7.85 (2.23)	1–10
*COVID-19 pandemic factors*			
Caregivers’ motivations of COVID-19 vaccination uptake			
Values		17.92 (2.81)	7–22
Impacts		16.88 (2.91)	5–21
Knowledge		14.88 (3.00)	6–21
Autonomy		14.00 (3.56)	4–21
Caregivers’ risk perception regarding COVID-19		14.02 (5.19)	1–28
Caregivers’ difficulty in managing their children’s self-protective behaviors		3.87 (3.56)	0–18
Caregivers’ mental health status during the COVID-19 pandemic			
Good	161 (63.9)		
Poor	91 (36.1)		
*Non-COVID-19 pandemic factors*			
Caregivers’ sex			
Female	200 (79.4)		
Male	52 (20.6)		
Caregivers’ age (years)		42.23 (8.34)	23–77
Caregivers’ years of education (years)		14.17 (2.67)	6–20
Caregivers’ parenting styles			
Affection/care		37.08 (5.16)	19–47
Overprotection		13.75 (3.32)	7–22
Authoritarian		12.24 (2.67)	6–21
Child’s sex			
Girl	52 (20.6)		
Boy	200 (79.4)		
Child’s age (years)		9.61 (2.39)	6–17
Child’s taking medication for ADHD			
No or irregular	40 (15.9)		
Regular	212 (84.1)		
Child’s inattention symptoms		12.88 (5.83)	0–27
Child’s hyperactivity/impulsivity symptoms		9.93 (6.17)	0–27
Child’s oppositional defiant disorder symptoms		9.33 (5.92)	0–23

ADHD: attention-deficit/hyperactivity disorder.

**Table 2 vaccines-10-00753-t002:** The association between the COVID-19 pandemic and non-COVID-19 pandemic factors related to the level of caregivers’ intentions to vaccinate their children with ADHD against COVID-19: univariate and stepwise multivariate regression analyses.

	Level of Intentions to Vaccinate Their Children	
	Univariate Regression	Stepwise Multivariate Regression
	B (SE)	B (SE)
*COVID-19 pandemic factors*		
Drivers of COVID-19 vaccination uptake		
Values	0.381 (0.045) ***	0.253 (0.063) ***
Impacts	0.339 (0.044) ***	0.124 (0.061) *
Knowledge	0.300 (0.044) ***	
Autonomy	0.177 (0.041) ***	0.105 (0.037) **
Risk perception regarding COVID-19	−0.037 (0.028)	
Difficulty in managing their children’s self-protective behaviors	−0.031 (0.040)	
Mental health status during the COVID-19 pandemic	−0.446 (0.297)	
*Non-COVID-19 pandemic factors*		
Caregivers’ sex	0.801 (0.350) *	0.716 (0.299) *
Caregivers’ age (years)	0.045 (0.017) **	
Caregivers’ years of education (years)	0.003 (0.054)	
Caregivers’ parenting styles		
Affection/care	0.025 (0.028)	
Overprotection	−0.004 (0.043)	
Authoritarian	−0.037 (0.054)	
Child’s sex	−1.043 (0.348) **	−0.760 (0.299) *
Child’s age	0.128 (0.060) *	0.112 (0.050) *
Child’s taking medication for ADHD	0.321 (0.392)	
Child’s inattention symptoms	−0.015 (0.025)	
Child’s hyperactivity/impulsivity symptoms	−0.032 (0.023)	
Child’s oppositional defiant disorder symptoms	−0.002 (0.024)	

ADHD: attention-deficit/deficit disorder; SE: standard error. *: *p* < 0.05; **: *p* < 0.01; and ***: *p* < 0.001.

## Data Availability

The data will be available upon reasonable request to the corresponding authors.

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
