# Peer review of "Second Wave of the Study of Taiwanese Caregivers of Children with ADHD in the COVID-19 Pandemic: Intentions to Vaccinate Their Children for COVID-19, and Related Factors"

_vaccines, 2022, doi:10.3390/vaccines10050753_

Round 1
Reviewer 1 Report
This article present a study of the attitude towards COVID-19 Vaccine among the Taiwanese Caregivers of ADHD children.
The title is apposite but probably too long and the keywords should be chosen more wisely (it's futile to have keywords all included in the title; for example, "ADHD" should be used instead of "Attention-Deficit/Hyperactivity Disorder" in title or keyword).
The abstract is a bit repetitive, I suggest to reduce the number of repeted words and concepts.
The introdution is sufficent to provide the necessary background and the references are relevant and recent enough (I think that the data reported at the beginning of Introduction are referred to April 2022 and not April 2021).
The methods and procedures applied are clearly described and reasonable.
Tables' readibility and labelling are generally good.
The discussion of data is clear and the conclusions seem consistent with them; , the limitations of the study are honestly described.
A general light revision of the English language is suggested to reduce some repetitions and improve the readability of some paragraphs.
The scientific soundness and significance of content are average good, the originality is not so high generally speaking, but it's enhanced by the particular focus chosen (caregivers of ADHD children), so this study may be of interest for the readers of the journal.
Everything considered, in my opinion this review is suited for publication (after suggested editings).
Author Response
We appreciated your valuable comments. As discussed below, we have revised our manuscript with underlines based on your suggestions. Please let us know if we need to provide anything else regarding this revision.
Comment 1
The title is apposite but probably too long and the keywords should be chosen more wisely (it's futile to have keywords all included in the title; for example, "ADHD" should be used instead of "Attention-Deficit/Hyperactivity Disorder" in title or keyword).
Response
Thank you for your comment. We replaced "Attention-Deficit/Hyperactivity Disorder" by “ADHD” in the title and key word. Please refer to line 2 and 39.
Comment 2
The abstract is a bit repetitive, I suggest to reduce the number of repeted words and concepts.
Response
We reduced the number of repeated words and concepts in Abstract. Please refer to line 18-38.
Comment 3
The introdution is sufficent to provide the necessary background and the references are relevant and recent enough (I think that the data reported at the beginning of Introduction are referred to April 2022 and not April 2021).
Response
Thank you for your reminding. We corrected the error from “April 3, 2021” into “April 3, 2022.” Please refer to line 44.
Comment 4
The methods and procedures applied are clearly described and reasonable.
Response
Thank you for your positive comment.
Comment 5
Tables' readibility and labelling are generally good.
Response
Thank you for your positive comment.
Comment 5
The discussion of data is clear and the conclusions seem consistent with them; the limitations of the study are honestly described.
Response
Thank you for your positive comment.
Comment 6
A general light revision of the English language is suggested to reduce some repetitions and improve the readability of some paragraphs.
Response
We have invited an English native editor to revise the manuscript. Attached please find the certificate.
Comment 7
The scientific soundness and significance of content are average good, the originality is not so high generally speaking, but it's enhanced by the particular focus chosen (caregivers of ADHD children), so this study may be of interest for the readers of the journal.
Response
Thank you for your positive comment.
Comment 8
Everything considered, in my opinion this review is suited for publication (after suggested editings).
Response
Thank you for your positive comment.
Reviewer 2 Report
This is a paper evaluating the COVID-19 pandemic factors and non-COVID-19 pandemic factors influencing the intention to vaccinate children with Attention-Deficit/Hyperactivity Disorder. The paper is well-written and is of interest for the readers; However, several minor changes should be made before considering its publications.
The abstracts section does not adequately describe the method of the research questionnaire. Was this questionnaire designed by the authors? Is it structured or semistructured?
How is the impact of these factors? In the abstract section, in a last sentence, the authors should provide a brief perspective for the future. Is there any public health strategy to control these factors?
The introduction is well-written. I recommend adding several sentences introducing the COVID-19 pandemic before reporting data on the vaccination. A first step would be to introduce the covid-19 pandemic; as a second step, the vaccination in patients with mental health disorders; and finally, I would introduce the particularities in patients with ADHD.
The first paragraph of section 1.4. Study aims should be moved to the introduction section.
The subsection called "Aims" should be restricted to the information on the purpose of the study.
The methods section describe the participants and the study design, the measures and the COVID-19 pandemic factors and non-pandemic factors as well as assessment scales. This has been adequately described.
The discussion section is brief. I would expand the limitations and strengths of the study. I would consider to reinforce the idea that there is a paucity of studies investigating the vaccination in patients with mental disorders, particularly at this period of life.
Author Response
We appreciated your valuable comments. As discussed below, we have revised our manuscript with underlines based on your suggestions. Please let us know if we need to provide anything else regarding this revision.
Comment 1
The abstracts section does not adequately describe the method of the research questionnaire. Was this questionnaire designed by the authors? Is it structured or semistructured?
Response
Thank you for your comment. We revised the Abstract and added the names of the structured questionnaires that were well-developed by others researchers as below. Please refer to line 23-28.
“252 caregivers of children with ADHD completed the structured questionnaires, including the Drivers of COVID-19 Vaccination Acceptance Scale, Risk Perception of the COVID-19 Scale, Caregivers’ Difficulties in Asking Their Child to Adopt Self-Protective Behavior Scale, Brief Symptom Rating Scale, Parental Bonding Instrument, Swanson, Nolan, and Pelham, version IV Scale, and questionnaires for the intention to vaccinate their children and child’s medication use for treating ADHD.”
Comment 2
How is the impact of these factors? In the abstract section, in a last sentence, the authors should provide a brief perspective for the future. Is there any public health strategy to control these factors?
Response
Thank you for your comment. We revised the last sentence of the Abstract as below to considering the impacts of the related factors identified in this study. Please refer to line 36-38.
“As a modifiable factor, caregivers’ drivers of COVID-19 vaccination uptake should be the target of intervention. Intervention programs for enhancing caregivers’ intention should be specified according to sex and age of caregivers and children.”
Comment 3
The introduction is well-written. I recommend adding several sentences introducing the COVID-19 pandemic before reporting data on the vaccination. A first step would be to introduce the covid-19 pandemic; as a second step, the vaccination in patients with mental health disorders; and finally, I would introduce the particularities in patients with ADHD.
Response
- Thank you for your suggestion. In the revised manuscript we added a new paragraph “1.1. Coronavirus Disease in the General Population and Individuals with Mental Disorders” (line 42-62) to introduce COVID-19 pandemic in the general population, individuals with mental health disorders, and individuals with ADHD.
“Coronavirus disease (COVID-19) caused by severe acute respiratory syndrome coronavirus 2 (SARS-CoV-2) has spread worldwide. As of April 3, 2022, the World Health Organization (WHO) had recorded 486,761,597 confirmed COVID-19 cases including 6,142,735 deaths [1]. As a novel respiratory infectious disease that is highly contagious, the COVID-19 pandemic has not only impacted physical health [2] but also mental health [3,4], the economy [5], education [6], quality of life [7], occupations [8], and the interpersonal relationships [9] of humans.
The meta-analysis studies have found that individuals with pre-existing mental disorders are had increased risks of COVID-19 mortality [10,11] and hospitalization [11]. The results of meta-analysis studies highlight the need for targeted approaches to manage and prevent COVID-19, for example, vaccination in individuals with mental disorders [10,11]. Along with psychotic and mood disorders, attention-deficit/hyperactivity disorder (ADHD) has been identified to predict the outcomes of contracting COVID-19. Compared with individuals without ADHD, those with ADHD have a higher risk of contracting respiratory infectious diseases (RIDs) [12], including COVID-19 [13,14]. Moreover, ADHD is a risk factor for poor COVID-19 outcomes (e.g., poor adaptation to the pandemic, severe COVID-19 symptoms, and increased referral for hospitalization) [15,16]. Caregivers of children with ADHD reported experiencing heavy care burden, stressful feelings, mental health disturbances, and compromised quality of life during the COVID-19 pandemic [16-18].”
- Then we revised the contents of “1.2. Vaccines for COVID-19” (line 63) by introducing vaccines for COVID-19 in the general population, individuals with mental health disorders, and children with ADHD. Below we added the introduction for COVID-19 vaccination in individuals with mental disorders. Please refer to line 67-72.
“However, a study in Denmark found that vaccine willingness for preventing COVID-19 was lower amongst individuals with mental disorders compared with the general population [21]. A study in China found that individuals with mental disorders had a much lower rate of vaccination than the general population [22]. These studies have suggested that it is emergently necessary to develop strategies for increasing vaccination coverage among individuals with mental disorders [21,22].”
Comment 4
The first paragraph of section 1.4. Study aims should be moved to the introduction section. The subsection called "Aims" should be restricted to the information on the purpose of the study.
Response
Thank you for your suggestion. In the revised manuscript we moved the first paragraph of ”Study aims” into “1.5. Study on Taiwanese Caregivers of Children with ADHD in the COVID-19 Pandemic.” Please refer to line 136-163.
Comment 5
The methods section describe the participants and the study design, the measures and the COVID-19 pandemic factors and non-pandemic factors as well as assessment scales. This has been adequately described.
Response
Thank you for your positive comment.
Comment 6
The discussion section is brief. I would expand the limitations and strengths of the study. I would consider to reinforce the idea that there is a paucity of studies investigating the vaccination in patients with mental disorders, particularly at this period of life.
Response
Thank you for your comment. We expanded the section of “strengths and limitations” by added the contents below into the revised manuscript. Please refer to line 413-419.
“This study focused on the intention to vaccinate children against COVID-19 and related factors in caregivers of children with ADHD. The results of this study provided the references for developing intervention programs to enhance caregivers’ intention. Given that individuals with mental disorder are the group at the risk of contracting COVID-19 and poor outcomes [28,29], further study is needed to examine the real rate of vaccination, vaccine willingness, barriers to assess vaccination, and effects of vaccination for COVID-19 in individuals with mental disorders during the COVID-19 pandemic.”